# An All-Fiber Fabry–Pérot Sensor for Emulsion Concentration Measurements

**DOI:** 10.3390/s23041905

**Published:** 2023-02-08

**Authors:** Simon Pevec, Janez Kunavar, Vedran Budinski, Matej Njegovec, Denis Donlagic

**Affiliations:** 1Laboratory for Electro Optics and Sensor Systems, Faculty of Electrical Engineering and Computer Science, University of Maribor, Koroska Cesta 46, 2000 Maribor, Slovenia; 2Marovt d.o.o., Stranice 55, 3206 Stranice, Slovenia

**Keywords:** optical fiber sensors, Fabry–Pérot interferometer, refractive index, temperature, emulsion concentration

## Abstract

This paper describes a Fabry–Pérot sensor-based measuring system for measuring fluid composition in demanding industrial applications. The design of the sensor is based on a two-parametric sensor, which enables the simultaneous measurement of temperature and refractive index (RI). The system was tested under real industrial conditions, and enables temperature-compensated online measurement of emulsion concentration with a high resolution of 0.03 Brix. The measuring system was equipped with filtering of the emulsion and automatic cleaning of the sensor, which proved to be essential for successful implementation of a fiber optic *RI* sensor in machining emulsion monitoring applications.

## 1. Introduction

Fiber-optic refractive index (RI) sensors have a significant potential for characterization of liquids and gases [1] in a wide area of demanding industrial applications [2]. Small dimensions, a fully electrically passive and dielectric design, intrinsic explosion safety, compatibility with chemically aggressive environments, and high temperature operation capability are just a few distinctive advantages that make fiber optic *RI* sensors attractive for industrial use. The implementation of a fiber optic *RI* sensor in industrial environments, however, proves to be a challenging task, due to inevitable issues with contamination of the involved optical surfaces. Thus, while fiber optic *RI* sensors present one of the most intensively and widely investigated areas of fiber-optic sensor technology [3,4], as reported in a substantial number of scientific papers [1,5,6,7,8], their commercial availability and industrial use remains restricted [9].

Within this work we attempt to utilize an all-fiber Fabry–Pérot Interferometer with an open path microcell structure for the measurement of the *RI* of cooling and lubrication emulsion, which are used in the metal manufacturing industry. Cooling and lubrication emulsions, which are mostly fine dispersions of oil in water [10], are of special interest in subtractive machining processes. Oil-in-water emulsions combine the cooling properties of water with the lubricating properties of oil, and take care of chip removal and tool wear out. They are used regularly and frequently and in large quantities. The ratio between the water and oil content in an emulsion must be maintained actively to achieve optimum results. The emulsion concentration in the subtractive machining processes is, thus, an important parameter, which affects the machining process. The change in emulsion concentration is affected mainly by three mechanisms: water evaporation, the presence of impurities and contaminants, and sticking onto the workpieces and chips during the manufacturing process, which are removed from the process actively. Too much water in the emulsion results in a loss of lubrication properties, and too much oil in the emulsion can result in higher cutting temperatures and improper viscosity that can lead to decreased pumping efficiency and loss of emulsion delivery to critical locations. This can lead to damage of cutting tools, inserts, parts, and poor final quality of products. Online monitoring of emulsions is, thus, a key capability in automation and predictive maintenance in modern metal processing facilities.

Nowadays, manual sampling and reading of emulsion concentration is used for most machines in the subtractive manufacturing industry. Most often, the refractometry approach is used with a handheld optical instrument, which has a typical measurement range between 1.3200 and 1.3348 RIU, which corresponds to 0% to 10% (Brix scale). This manual approach is time consuming, and cannot be used in automatic real-time monitoring. Continuous online measurement of the emulsion concentration would, thus, simplify emulsion concentration maintenance (dosing) significantly. By keeping the emulsion concentration within a narrow range of tolerances, the production process can perform under constant and optimal cooling and lubrication conditions. This contributes to higher allowable feed rates, deeper cuts, better accuracy, reduced power consumption, longer tool life, and better and more repeatable final product quality and reduced consumption of emulsion.

An adequate compact *RI* sensor that can measure emulsion concentration by determining *RI* is, thus, of high interest to a broad range of subtractive machining processes in related industries.

## 2. Sensor Design

The proposed sensor design is presented in Figure 1, and depicts two Fabry–Pérot Interferometers (FPIs) with an open path microcell consisting of three in-fiber hermetically sealed mirrors. Mirrors 1 and 2 define a temperature sensing interferometer (discussed further in detail below), while mirrors 2 and 3 define a (partially) open-path FPI, which is used for *RI* determination. Open-path fiber-optic FPI sensors have one distinctive, but important advantage over most other fiber-optic *RI* designs: They do not rely on an optical field, which would be confined tightly to the surface of the waveguide. The latter always acts as an area prone to capturing of undesired contaminants and residues. Open path sensors, such as microcell FPIs [11], measure the *RI* of a small bulk volume of the fluid, and thus, always yield a substantial ratio between a sample and a potential undesired residue/contaminant volume, which provides a reasonable tolerance of the sensor to a potential increase in the volume of the contaminant. The open path FPI’s sensitivity to residue is, thus, generally decreased by increasing the length of the microcell optical path involved in the sensor design. In contrast, sensors that exploit evanescent field interactions (most FBG *RI* sensors [12,13,14], LPFG sensors [15,16,17], modal interference sensors [18,19,20], etc.) exhibit substantially higher sensitivity to the formation of deposits and residues, as even a very thin formation of the undesired residue film on the sensor surface results in a substantial change in the average *RI* of the volume that is involved in the measurement of the RI.

The implementation of an open path all-fiber Fabry–Pérot Interferometer for a liquid *RI* measurement requires the formation of sufficiently reflective mirrors, which can provide robust long-term performance under harsh conditions. While a physical deposition of a thin oxide, other dielectric, or even metal layers, can be applied onto the surfaces of optical fibers to achieve nearly arbitrary reflectance [21], these types of mirrors mostly do not withstand aggressive chemical cleaning, or lack adequate long-term stability under harsh conditions. In the proposed design, we thus implemented hermetically enclosed in-fiber mirrors, which are not exposed directly to the surrounding liquid, and thus, provide good compatibility with chemically aggressive environments, including aggressive cleaning procedures using solvents, acids, heated acids, pyrolytic methods, and similar approaches. The compatibility of the *RI* sensor with aggressive cleaning is of special interest when the sensor is intended for use in contamination forming environments, as is the case with emulsions used in subtractive material processing. Even in a clean and filtered emulsion, solidified submicron layers of fine deposits will be formed over time, which causes additional optical losses and long-term *RI* measurement drifts associated with optical path length changes due to the gradual formation of the contamination layer.

Finally, temperature compensation must also be added, as all liquids exhibit high and nonlinear *RI* temperature dependence, which interferes with the emulsion concentration measurements. Temperature sensing is achieved by “sandwiching” of a fiber section between mirrors 1 and 2 (Figure 1), which forms a temperature sensing FPI. The temperature dependent change of the optical path length of this interferometer is caused predominately by a change of the fiber’s core RI. The proximity of the temperature and *RI* sensing regions ensures minimum temperature gradients, and thus, effective temperature compensation.

The *RI* measuring FPI is defined by mirrors 2 and 3, and is made of an open-path microcell, which is elongated further on both sides by a pair of short segments of standard single mode fibers. As the sensor is intended for use in liquids, the perpendicular surfaces on both sides of the microcell do not generate significant reflections, at the order of 0.2% (at the most), which simplifies signal interrogation. Fiber extension of the microcell acts primarily as a means to encapsulate the mirrors fully within the fiber, and to protect them from direct contact with the surrounding fluid.

When designing the sensor, the main goal was to measure the absolute refractive index of the emulsion with a sufficiently high resolution. Implementation of the frequency measurement algorithm (described in Section 5) required the longest possible microcell, as an increase the cell length decreased the *FSR* of the cavity, which further increased the number of special fringes that appeared within the given spectral range interval of the integrator, and consequently, allowed for more accurate determination of the spectral fringe frequency. The microcell length is, however, limited by the diffraction/beam divergence, scattering, and absorption losses. Initial measurements indicated significant light scattering in the emulsion, which increases the optical losses within the cavity. The measured losses at a typical emulsion concentration (*RI* = 5 Brix) were about 12 dB/mm. To achieve fringe contrast in the range of at least 20% while assuming both mirrors are about 4% reflective, we calculated that the length of the cells should not exceed 0.7 mm (the double length of the micro cell was considered for the calculation in the reflective mode). Considering the possibility of additional losses caused by contamination (absorption losses) in a real environment, we produced sensors with typical lengths (L_cell_) between 400 and 500 mm.

## 3. Sensor Production Process

The sensor manufacturing process consists of multiple splices, cleaves, and femtosecond laser micromachining steps. The entire process is shown in detail in Figure 2. The sensor manufacturing begins with the fabrication of the first in-fiber mirror, which is produced by splicing a single-mode optical fiber (SMF) sputtered with TiO_2_ to the lead-in SMF, as depicted in Figure 2a. All subsequent mirrors were also fabricated by TiO_2_ sputtering and the splicing process, as described in Reference [22]. The SMF length is then trimmed to a length of about 400 µm (Figure 2b), which also defines the FPI cavity length of the temperature sensor roughly. The temperature sensor is completed with the second in-fiber mirror, produced by another splice, as depicted in Figure 2c. In the next step, the spliced fiber is cleaved approximately 150 µm away (Figure 2d) from the second mirror, to provide a safe distance for the mirror, which can be damaged in the following splice (Figure 2e), where a capillary (Cap, ID/OD = 85/125 µm) is spliced to the pre-made assembly. The capillary length is then trimmed to a length which defined the active sensing area of the *RI* sensor (Figure 2f), and was, in our case, around 430 µm. The resulting assembly is then spliced to an SMF (Figure 2g), and cleaved into an approximately 150 µm long segment. The last splice is made to produce the third in-fiber mirror (Figure 2i), which, in combination with the first in-fiber mirror, forms the FPI, which will be used to measure the *RI* of the surrounding media. The end of the fiber was cut/broken roughly to give negligible reflection from the end surface of the end-cap. At the final production step, presented in Figure 2j, the capillary section is micromachined precisely by a femtosecond laser to gain a large square opening so that liquid can circulate through it easily.

## 4. Experimental Setup

The experimental setup as shown in Figure 3a consisted of the proposed sensor, a spectral interrogator, Micron Optics si155 (MO-Si155), a computer, a motorized XY stage, a container with emulsion (ZET-cut 9800) connected via pumps and filters to a large tank used by a CNC machine, and three cleaning agent containers filled with distilled water (H_2_O), isopropyl alcohol (IPA), and sulfuric acid (H_2_SO_4_).

The proposed sensor was connected to a spectral interrogator and attached to a motorized XY stage. The emulsion *RI* is measured in the container, where the emulsion is pumped continuously with a peristaltic pump through a filter of magnetic particles (Eclipse Magnetics MM5) and filters of any particles larger than 5 µm. With the XY stage, the sensor is transferred automatically through various cleaning processes with different sequences and time intervals. As part of the research, we found that the most effective method for cleaning greasy deposits on the sensor is by using H_2_SO_4_ heated to 120 °C. Due to the fast contamination of the acid when the sensor was transferred directly from the emulsion to H_2_SO_4_, we added pre-cleaning of the sensor, which is performed primarily in IPA and H_2_O. After the measurement of the *RI* was completed, the sensor was submerged into the IPA, then from the IPA it went into the H_2_O, and from the H_2_O into the H_2_SO_4_. From the H_2_SO_4_, the sensor went back to the IPA, then to the H_2_O, and from the H_2_O back to the emulsion, and the loop was completed.

## 5. Signal Interrogation

The signal interrogation was performed by using a spectral interrogator (MO-Si155), which reads the sensor’s back-reflected optical spectrum in the range from 1460 to 1620 nm. The interrogator reads the reflected optical spectra in 20,000 points and sends it over the LAN (TCP/IP) to a personal computer with an acquisition rate of 10 Hz. The data from the interrogator were then processed by a LabView software package. The acquired back-reflected optical spectra was first converted from the (original) wavelength to a frequency domain (Figure 4a). This assured that the spectral fringes were fully periodic in the acquired spectral data, i.e., the free spectral range (*FSR*—Figure 4d) was constant in the domain of optical frequencies (as opposed to the original wavelength domain, where the *FSR* increases with the increasing wavelength). After the conversion of the back-reflected optical spectrum into the frequency domain (Figure 4a), an Inverse Discrete Fast Fourier transform (IDFFT) was performed on the acquired spectral data (Figure 4b). The peak position in the IDFFT represents a round trip time of flight in the FP cavity, which we converted to the FPI optical length by multiplying the x-axis by c/2. Both FPI lengths of interest are visible in Figure 4b, as the sensor was immersed in water. Tracking of the phase(s) of the IDFFT components that correspond to the peak values in the absolute IDFFT allowed for efficient and high-resolution tracking of the FPI’s length changes, which was explained in detail in [23]. In the case of the proposed sensor, high-resolution tracking of the FPI’s length changes was used only for temperature reading, where the phase change/shift of the spectral fringe generated by the temperature sensing interferometer was correlated to a temperature change as follows:(1)$$\Delta T=\frac{\Delta \varphi_{T}\cdot \lambda }{4\pi \frac{dn_{SiO_{2}}}{dT}\cdot L}$$
where Δ*φ_T_* represents the phase shift of the temperature sensing FPI spectral fringe, which was calculated as the phase of the complex IDFFT component corresponding to the length of the temperature sensing segment, *λ* is the central wavelength of the signal interrogator, *dn*_*SiO*2_/*dT* is a thermo-optic coefficient for the silica fiber, and *L* is the length of the temperature related FPI. The spectral sensitivity of the temperature sensor in our case was 10.2 pm/°C.

However, the method based on phase tracking of the spectral fringe is not suitable for measuring the absolute *RI* of emulsion, because of ambiguity in the phase over a wide measurement range of RI, which changes for many multiples of 2π when the *RI* changes between 0 to 10 Brix (1.32 to 1.335 RIU). The spectral sensitivity of the *RI* sensor immersed in a typical emulsion (5 Brix) is approximately 600 nm/RIU. Therefore, a simple method was implemented for absolute measurement of the interferometers’ optical path length. For this purpose, the reflected spectrum in the frequency domain was first filtered with a high-order band pass filter (Figure 4c) with the central frequency that corresponded to an *RI* measurement interferometer spectral fringe frequency. Then, the optical frequency of the filtered signal was measured by using a LabView function for the measurement of a frequency within a data array. The inverse value of the measured frequency corresponded to the *FSR* (Figure 4d) of the *RI* sensing interferometer, and the *FSR* can be correlated further and directly to the *RI* of the fluid which fills the microcell:(2)$$FSR=\frac{c}{2nL}=\frac{c}{2\left[n_{SiO2}\left(L_{1}+L_{2}\right)+n_{fluid}L_{cell}\right]}$$
where *c* is the speed of light in the vacuum, *n* is an average refractive index, *L* is the length of an arbitrary FPI, *n_fluid_* is the refractive index of the fluid within the microcell, *n_SiO*2*_* is the refractive index of the silica fiber, *L_cell_* is the length of the microcell, and *L*_1_ and *L*_2_ are the lengths of both SiO_2_ sections on each side of the microcell, as presented in Figure 1. Thus, the refractive index of the fluid within the microcell can be calculated from Equation (2) as follows:(3)$$n_{fluid}=\frac{c-2\cdot FSR\cdot n_{SiO2}\left(L_{1}+L_{2}\right)}{2\cdot FSR\cdot L_{cell}}$$

The *RI* of a liquid is also affected strongly by the temperature. For example, a change in temperature of 1 K in the case of water will already cause a water *RI* change within the range of −1.1 × 10^−4^ RIU [24]. An *RI* sensor used for liquid composition analysis on a water basis and with a useful resolution in the range of 10^−4^ RIU or higher, thus, requires a temperature compensation. The temperature of the measuring *FPI_RI* (Figure 1) shall, thus, be measured with an accuracy, or at least repeatability, that is better than 1 °C. In the case of measuring the emulsion concentration, changes in temperature greater than 30 °C are not expected, which means that the phase change of the sinusoidal spectral component will be significantly smaller than 2π, meaning that the high-resolution method based on the phase tracking algorithm depicted above can be used for determining temperature. The measured temperature is applied to compensation of the *RI* measurement. The temperature sensitivity of the proposed *FPI_RI* arises from several factors: refractive index changes of silica fibers (*RI* change of segments *L*_1_ and *L*_2_), refractive index change of the measured liquid, and from the thermal expansion of segments *L*_1_ and *L*_2_. Due to the low temperature expansion of the SiO_2_, which is in the range of 5 × 10^−7^ K^−1^, contributions related to *RI* changes dominate. Therefore, we can assume that the temperature sensitivity of the *RI* measurement arises mainly from the change in the *RI* of SiO_2_ and the liquid. Taking this into account, we can rewrite Equation (2) as follows:(4)$$\begin{array}{l} FSR=\frac{c}{2\left[n_{SiO_{2}}\left(L_{1}+L_{2}\right)+\frac{dn_{SiO_{2}}}{dT}\cdot \left(L_{1}+L_{2}\right)\cdot \Delta T+n_{fluid}\cdot L_{cell}+\frac{dn_{fluid}}{dT}\cdot L_{cell}\cdot \Delta T\right]}=\\ =\frac{c}{2\left[n_{SiO_{2}}\left(L_{1}+L_{2}\right)+n_{fluid}\cdot L_{cell}+\left(\frac{dn_{SiO_{2}}}{dT}\cdot \left(L_{1}+L_{2}\right)+\frac{dn_{fluid}}{dT}\cdot L_{cell}\right)\cdot \Delta T\right]} \end{array}$$

From which it follows that the temperature compensated *RI* can be calculated as:(5)$$\begin{array}{l} n_{fluid}=\frac{c}{2\cdot FSR\cdot L_{cell}}-\frac{n_{SiO_{2}}\left(L_{1}+L_{2}\right)}{L_{cell}}-\Delta T\left[\frac{\frac{dn_{SiO_{2}}}{dT}\cdot \left(L_{1}+L_{2}\right)}{L_{cell}}+\frac{dn_{fluid}}{dT}\right] \end{array}$$

For a better presentation of Equation (5), it can be written by introducing coefficients *K*_1_, *K*_2_, and *K*_3_ as follows:(6)$$n_{fluid}=\frac{K_{1}}{FSR}-\Delta T\cdot \left[K_{2}+\frac{dn_{fluid}}{dT}\right]-K_{3}$$
where
(7)$$K_{1}=\frac{c}{2L_{cell}};\;K_{2}=\frac{\frac{dn_{SiO_{2}}}{dT}\cdot \left(L_{1}+L_{2}\right)}{L_{cell}};\;K_{3}=\frac{n_{SiO_{2}}\left(L_{1}+L_{2}\right)}{L_{cell}}$$

While a linear relationship between the *RI* change of the silica fiber and temperature can be assumed (i.e., *dn*/*dT* for SiO_2_ is constant and in the range of 10^−5^/°C), a change of the fluid’s *RI* versus temperature deviates from the linear relationship. Since the emulsion is made on a water basis, we can expect that *dn*/*dT* will also be in the range of −1.1 × 10^−4^/°C [24], but for a precise determination, *dn*/*dT* versus temperature needs to be determined experimentally by an appropriate calibration procedure. This was achieved by immersing the sensor in an emulsion of known concentration (5.2 Brix), heating it slowly and recording the temperature and the *FSR* simultaneously. The obtained results were then used to calculate *dn*/*dT* according to Equation (6) over the temperature range of interest. The results are presented in Figure 5, and indicate an approximately constant value of *dn*/*dT* over the temperatures of interest. We can see that, compared to pure water, the *dn*/*dT* is higher, and at a concentration of 5.2 Brix, the *dn*/*dT* corresponds approximately to −1.5 × 10^−4^/°C.

## 6. Experimental Results and Analysis

The static characteristic of the produced sensor was measured by immersing the sensor in an emulsion from 4 to 8 Brix with a step of 0.5 Brix, and distilled H_2_O for reference. We chose this range as, for the cooling lubricant ZET-cut 9800, this is the standard working range used for grinding and machining. We repeated the measurements six times, and calculated the Standard Deviation at each measuring point, as shown in Figure 6. After each measuring point, the sensor was cleaned in heated H_2_SO_4_, and submerged in IPA and H_2_O.

With the following experiment, we showed the importance of temperature compensation. We prepared a larger amount of emulsion with a value of *RI* = 5.2 Brix. The emulsion was then distributed between several containers, which were stabilized at different temperatures. Then, the proposed sensor was used to measure the temperature and *RI* simultaneously in each of the heated containers. The graph in Figure 7 shows the uncompensated value of the *RI* calculated by Equation (3), the temperature compensated value of the *RI* calculated by Equation (6), and the temperature extracted from Δ*φ_T_* using Equation (1). The results confirmed the success of the temperature compensation, as the temperature-compensated *RI* values showed an approximately constant value in a temperature range from 25 to 45 °C.

The next experiment investigated the resolution of the *RI* sensing system experimentally. Two emulsion concentrations with values of 5.00 Brix and 5.03 Brix were prepared in two 100 ml containers. The experiment was carried out at room temperature, by moving the sensor manually from one container to another while measuring the refractive index n_fluid_ according to Equation (6). The results presented in Figure 8 were recorded at an acquisition rate of 0.5 Hz, and show the ability of the sensor system to measure emulsion concentration with a resolution of at least 0.03 Brix or 4.45 × 10^−5^ RIU.

A series of initial tests were conducted to identify the drifts present in the long-term measurement of the *RI* of the emulsion. A typical test is shown in Figure 9, where we exposed the sensor continuously to emulsion for several days. The sensor drifted continuously (the composition of the emulsion was not changed during this test and corresponded to about 6 Brix). The drift in the *RI* measurement can be explained by the formation of solidified nanolayers that accumulated on all surfaces which were exposed to the emulsion. It should be stressed that the emulsion was designed to provide good wetting and adhesion to most materials, as these properties are required for good lubrication that is one of the main functions of the emulsion in CNC machining [25]. Periodic cleaning of the sensor was introduced to provide consistent long-term performance. This was done by automatic cleaning of the sensor in H_2_SO_4_ heated to 120 °C in 24 h intervals. As shown by Figure 9, the measured n_fluid_ was restored to its initial value successfully after each cleaning cycle. We also tested several alternative cleaning agents, including solvents such as acetone, isopropyl alcohol, and tetrahydrofuran, and also the pyrolytic technique, but none of the methods achieved results comparable to heated H_2_SO_4_.

After the initial laboratory test phase, the complete measurement system (shown in Figure 3) was transferred to an industrial environment, where we monitored the emulsion concentration in the case of an operating CNC machine. Due to the rapid deposition of contamination layers on the fiber surfaces, we initiated the online emulsion concentration measurement procedure, which consisted of semi-cleaning measurement and full cleaning cycles. The semi-cleaning measurement cycle included the following steps: The sensor was exposed to the emulsion for 1 min while measuring *RI* and temperature data; then, the sensor was transferred to the IPA for 8 min, and finally, the sensor was transferred from the IPA to water for 1 min. This 10 min cycle was repeated continuously to obtain continuous data on *RI* in the system. This semi-cleaning measurement cycle was interrupted in 8 h intervals, when the sensor cleaning was initiated using heated H_2_SO_4_. The cleaning cycle in H_2_SO_4_ lasted 5 min, and started by moving the sensor from the water to the H_2_SO_4_, which was heated to 120 °C. After cleaning the sensor in H_2_SO_4_, the sensor was transferred to the IPA (8 min), from there to the H_2_O (1 min), and then back to the emulsion (1 min) to restart the semi-cleaning measurement cycle. In the 8-hour cleaning cycle, cleaning in hot H_2_SO_4_ limited the measurement errors to below ±0.1 Brix. The entire sequence is presented in Figure 10. This procedure, while perhaps not optimized fully, was determined by substantial experiments, where we tested out different possible cleaning scenarios. We targeted the use of minimum quantities of cleaning agents (IPA and H_2_SO_4_) to minimize the waste generated during the sensor operation. We also found that frequent exposure of the sensor to H_2_SO_4,_ results in a relatively rapid contamination of the acid, which results in the formation of stubborn residues on the sensor, while frequent cleaning with IPA prolonged the time interval that requires an aggressive hot acid cleaning interval, and reduced the acid contamination significantly. Additionally, reducing the sensor exposure time to the minimum required time reduced the amount of residue formation on the sensor.

The results in Figure 11 show the online measurements of the *RI* of emulsion in an industrial plant over a period of more than 2 weeks. The figure shows a comparison between the calculated values of the *RI* and the readings that are taken manually once per day using a handheld refractometer (RHB-10ATC). The results show a very good match in all measurement points. The more significant deviation between the expected value is related to the machine stops during the weekend when fresh emulsion was no longer supplied to the measurement site, which might also be a realistic situation due to the absence of emulsion circulation within the machining system.

## 7. Conclusions

This paper presented an all-fiber Fabry–Pérot sensor-based measuring system for measurement of the concentration of emulsion, which is used in subtractive machining of metals and other materials. Emulsion *RI* measurements are of special interest in industrial processes, since they provide one of the most direct methods for determination of emulsion concentration and overall emulsion quality. Emulsions, on the other hand, form strong and pronounced contamination layers/residues quickly and profoundly, which prevent the use of most *RI* sensors over longer periods of time. Therefore, we adopted a sensing system that incorporated a continuous and sufficiently efficient/aggressive cleaning procedure. To make the *RI* sensor compatible with the aggressive cleaning procedures, the proposed sensor design was based on hermetically sealed in-fiber mirrors, which provided compatibility with a wide range of cleaning agents, such as various solvents, acids, heated acids, pyrolytic techniques, and others. The emulsion concentration was also determined by measuring the *RI* and temperature simultaneously on the same optical fiber, which also enabled efficient temperature compensation.

The proposed sensor system achieved absolute *RI* measurements’ resolution in the range of 4.5 × 10^−5^ RIU, or 0.03 Brix, which exceeded the needs for emulsion concentration measurements. The small dimension of the sensor allowed the design of a simple system for continuous maintenance of the sensor’s optical surfaces by adaptation of the automatic cleaning protocol, which used small quantities of acids and solvents. The proposed sensor was demonstrated successfully in real-time, in situ measurements of emulsion concentration in an industrial environment (in an operating CNC machine).

## Figures and Tables

**Figure 1 sensors-23-01905-f001:**
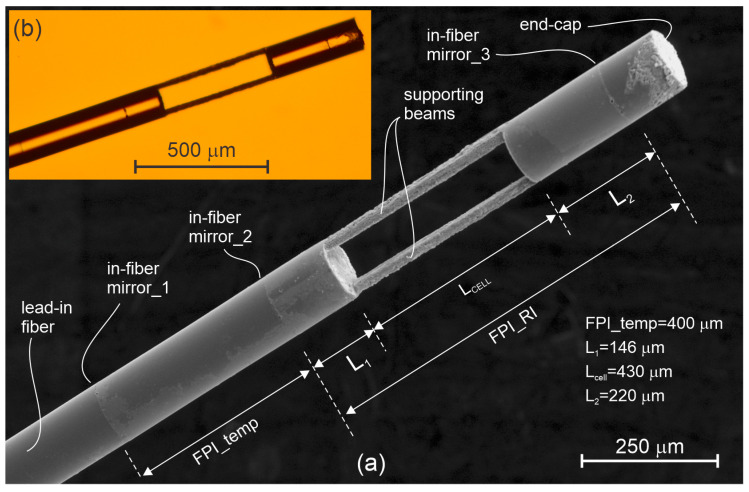
Produced sensor. (**a**) SEM image of a sensor: L_1_ = 146 μm, L_cell_ = 430 μm, and L_2_ = 220 μm. (**b**) Optical microscopic image of a sensor.

**Figure 2 sensors-23-01905-f002:**
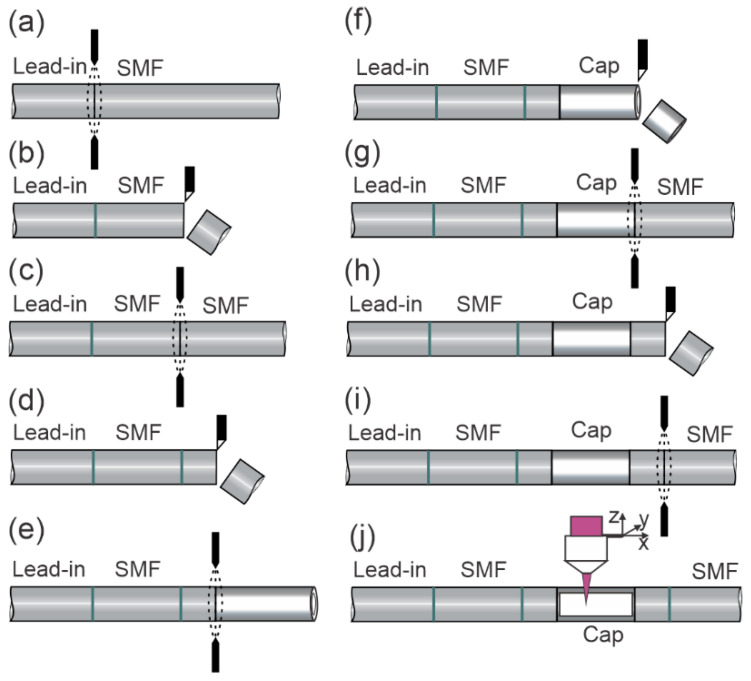
Production process: (**a**) fiber splicing–fabrication of the 1st in-fiber mirror, (**b**) precision fiber cleaving, (**c**) fiber splicing–fabrication of the 2nd in-fiber mirror, (**d**) fiber cleaving, (**e**) splicing capillary to the pre-made assembly, (**f**) precision capillary cleaving, (**g**) splicing SMF to pre-made assembly, (**h**) SMF cleaving into at least 150 μm long segment, (**i**) fiber splicing–fabrication of the 3^rd^ in-fiber mirror, (**j**) femtosecond laser micromachining.

**Figure 3 sensors-23-01905-f003:**
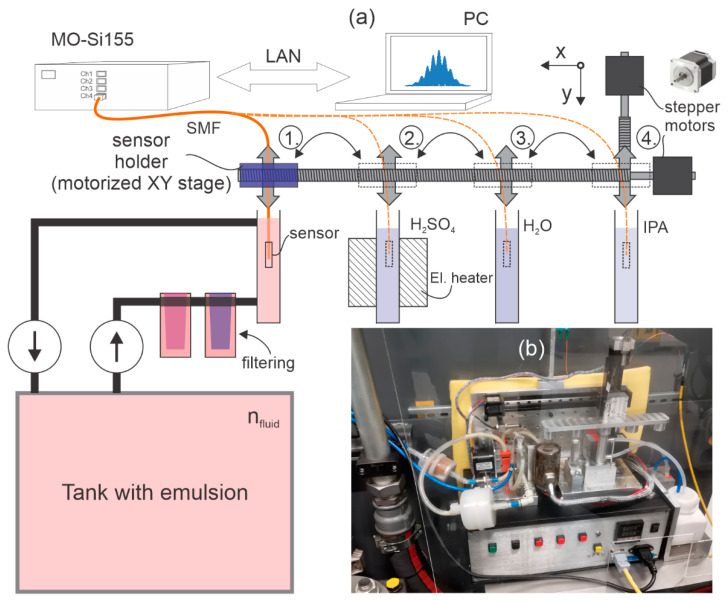
(**a**) Schematic of an online emulsion concentration meter for an industrial environment. The system consists of: (1) A measurement container (which is further connected to the main CNC machine emulsion reservoir (n_fluid_ ≈ 5.5 Brix) through a set of filters and pumps to allow circulation of the measured emulsion); (2) An H_2_S0_4_ container; (3) An H_2_O container and (4) An IPA container. All four containers had a volume of about 7.5 ml. A compact set of x-y linear translation stages provided programmable movement of the sensor between the containers. The sequence of sensor movements is automatic and fully programmable. (**b**) Mechanical part with motorized XY stage, containers, filters, and pumps installed in an industrial environment in the CNC machine.

**Figure 4 sensors-23-01905-f004:**
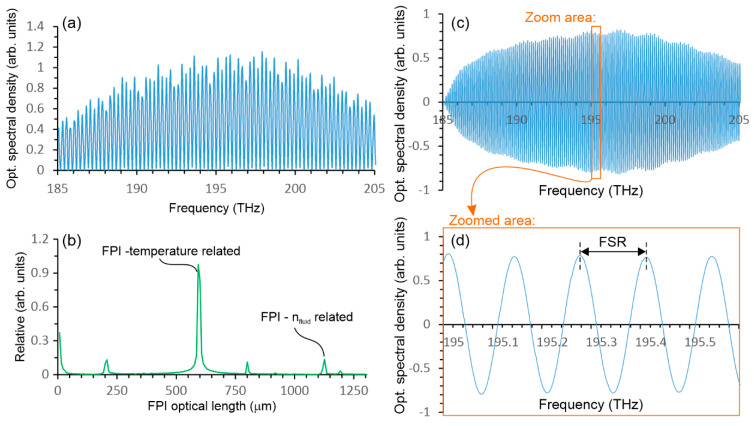
(**a**) The sensor’s back reflected the optical spectrum. (**b**) Absolute IDFFT with the x axis converted into an optical FP cavity length. (**c**) The sensor’s back reflected spectrum filtered by a bandpass filter to pass only the spectrum component related to the n_fluid_-related FPI. (**d**) Magnified small area of the filtered back reflected spectrum.

**Figure 5 sensors-23-01905-f005:**
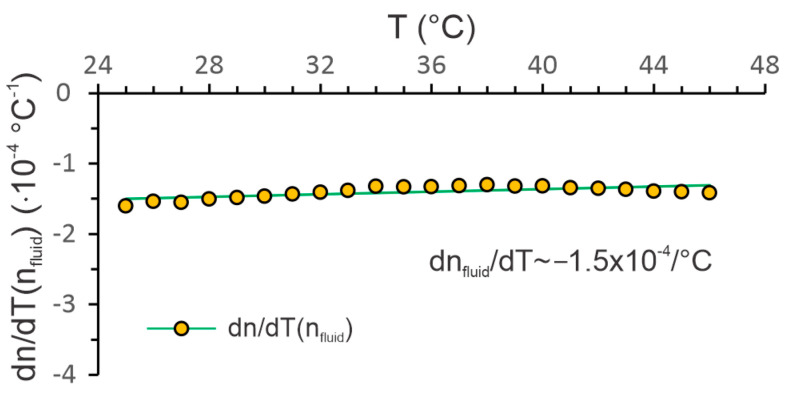
Measured *dn*/*dT* of the target emulsion at n_fluid_ = 5.2 Brix.

**Figure 6 sensors-23-01905-f006:**
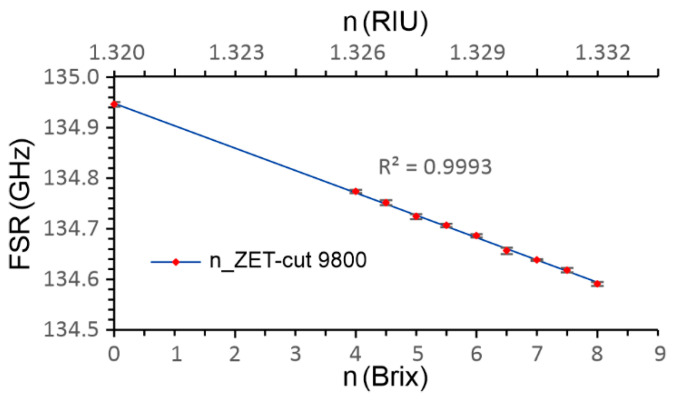
Static characteristic for an *RI* sensor measured in the prescribed working range for emulsion from 4 to 8 Brix, using distilled water as the reference point.

**Figure 7 sensors-23-01905-f007:**
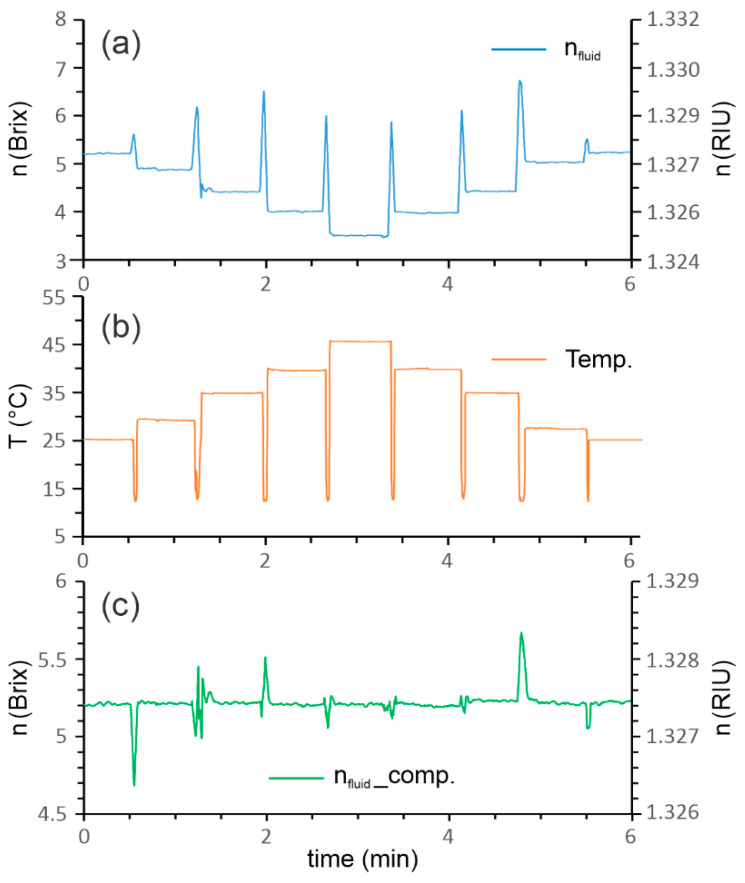
Refractive index measurement of the typical emulsion (approximately 5.2 Brix) at different points of temperature: (**a**) uncompensated *RI*, (**b**) temperature of the emulsion, and (**c**) temperature compensated *RI*.

**Figure 8 sensors-23-01905-f008:**
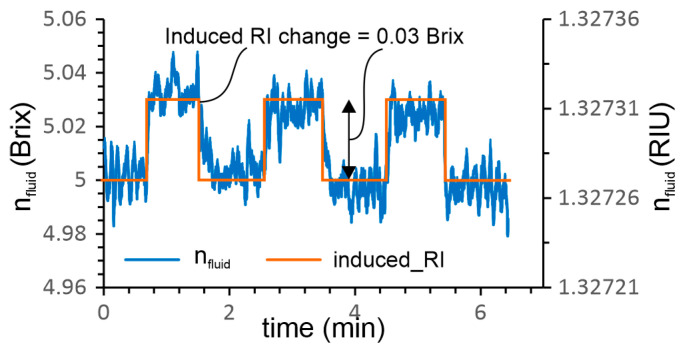
Estimation of the sensing systems’ *RI* resolution.

**Figure 9 sensors-23-01905-f009:**
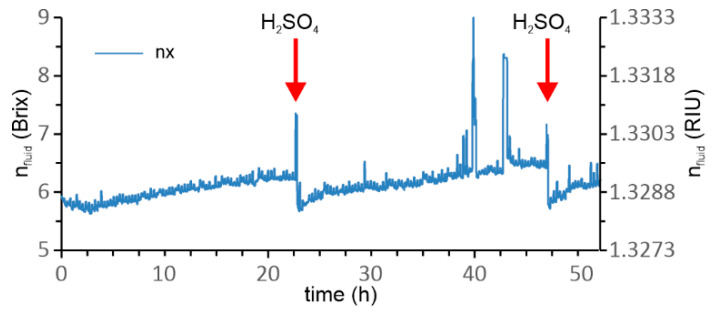
Refractive index measurements for 50 h with 24 h interval cleaning in H_2_SO_4_.

**Figure 10 sensors-23-01905-f010:**
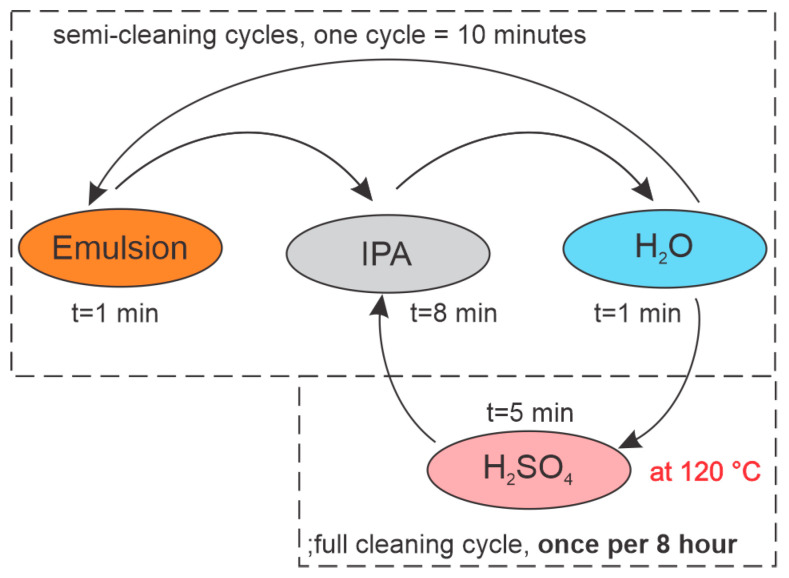
The time sequence of measuring performed in an industrial environment.

**Figure 11 sensors-23-01905-f011:**
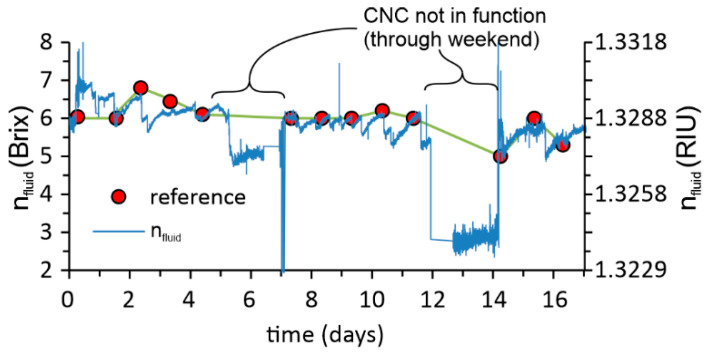
Comparison between the manual *RI* measurements using a handheld refractometer RHB-10ATC (one point per day) and our presented method in sixteen days.

## Data Availability

Not applicable.

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
