# Peer review of "An All-Fiber Fabry–Pérot Sensor for Emulsion Concentration Measurements"

_sensors, 2023, doi:10.3390/s23041905_

Round 1

Reviewer 1 Report

The authors present a Fabry-Perot sensor capable of detecting temperature and refractive index simultaneously and in real time. The developed sensor was tested in the laboratory environment and implemented in real-life work conditions by adding a self-cleaning system. The manuscript is well-written, and clear, with high-quality graphics. Some questions and comments on the work are following.

- On page 8, the authors present Equation 5 and the following equation is numbered 7. Is this a typo or an equation is missing?

- On page 8, the coefficients K1,2,3 are defined but not used. Are they relevant?

- On page 8 line 255, and page 9 line 290, the authors mention equation 6 but it doesn't exist.

- The graph in Figure 7 is not very clear. What is the purpose of the arrows of the lines corresponding to nfluid and nfluid_comp? A suggestion for the presentation of the data: do 3 plots while maintaining a common x-axis. It may show the simultaneous dips of nfluid and temperature, and allows the reader to see the almost continuous trend on nfluid_comp.

- On page 10, line 295. The phrase "A series of different tests (...) to establish the stability and possibility to maintain stability of the sensor (...)." is not clear.

- In the production process, the authors should make clear if all mirrors have TiO2.

- Is the end-cap cleaved orthogonally to the propagating light or does it have any angle? Is the back-reflected light at this surface taken into account? Does it have any significance on the processed signal?

- What is the error of the readings after 8h, before redoing the cleaning process?

Reviewer 2 Report

1.Please change Fig.3 to the practical experiment picture
2.Adding the sensor design simulation and optimization of sensor parameters are nice to prove your sensor designing idea.

3.The whole english writing needs to be improved. 

Reviewer 3 Report

Article:

An all-fiber Fabry-Perot sensor for emulsion concentration measurements

The authors present an article about An all-fiber Fabry-Perot sensor for emulsion concentration measurements

This is a well-written and structured manuscript. In this manuscript there are the potentialities to publications in Sensors but the paper has needed some important specification and reviews.

In the paper the state of art about the optical fiber sensor can be improved with more specifics and recently papers.

In particular:

Sentence:

 Thus, while fiber optic RI sensors present one of the 28 most intensively and widely investigated areas of fiber-optic sensor technology [3, 4], reported in a substantial number of scientific papers [1, 5], their commercial availability and 30 industrial use remains restricted [6].

* reported in a substantial number of scientific papers [1, 5].

Interesting and more recent papers, which have to adding, in this field are: Design of highly sensitive interferometric sensors based on subwavelength grating waveguides operating at the dispersion turning point, Journal of the Optical Society of America B, 2021, Fuentes at al. Improving the width of lossy mode resonances in reflection configuration D-shaped fiber by nanocoating laser ablation, Optilca letters, 2020, Paladino, D. et al. Hybrid fiber grating cavity for multi-parametric sensing. Opt. Expr. 2010.

In particular, regarding the Paladino, D. et al., the authors can be comparing the realized device with the interesting device shown in the submitted paper.

- The label T(°C) of figure 5 have been located in the upper part of plot.

- The label and axis n(Brix) of figure 6 have been located in the upper part of plot.

The authors present the sensor in all aspect about the realization and the application but nothing about the SRI sensitivity or temperature sensitivity. There parameters are important for a comparison with the state of art.

Round 2

Reviewer 3 Report

The paper can be accepted in the present form